# Fms-like tyrosine kinase 3 is a regulator of the cardiac side population in mice

Giacomo Della Verde[1],*, Michika Mochizuki[1],*, Vera Lorenz[1], Julien Roux[1,2], Lifen Xu[1], Leandra Ramin-Wright[1], Otmar Pfister[1,3],†, Gabriela M Kuster[1,3],†

Fms-like tyrosine kinase 3 (Flt3) is a regulator of hematopoietic progenitor cells and a target of tyrosine kinase inhibitors. Flt3-targeting tyrosine kinase inhibitors can have cardiovascular side effects. Flt3 and its ligand (Flt3L) are expressed in the heart, but little is known about their physiological functions. Here, we show that cardiac side population progenitor cells (SP-CPCs) from mice produce and are responsive to Flt3L. Compared with wild-type, flt3L$^{-/-}$ mice have less SP-CPCs with less contribution of CD45$^-$CD34$^+$ cells and lower expression of genes related to epithelial-to-mesenchymal transition, cardiovascular development and stem cell differentiation. Upon culturing, flt3L$^{-/-}$ SP-CPCs show increased proliferation and less vasculogenic commitment, whereas Akt phosphorylation is lower. Notably, proliferation and differentiation can be partially restored towards wild-type levels in the presence of alternative receptor tyrosine kinase-activating growth factors signaling through Akt. The lower vasculogenic potential of flt3L$^{-/-}$ SP-CPCs reflects in decreased microvascularisation and lower systolic function of flt3L$^{-/-}$ hearts. Thus, Flt3 regulates phenotype and function of murine SP-CPCs and contributes to cellular and molecular properties that are relevant for their cardiovasculogenic potential.

## Introduction

Tyrosine kinase inhibitors (TKIs) used for leukemia and solid tumor treatment can have cardiovascular side effects, which include cardiac dysfunction and heart failure (Pun & Neilan, 2016). The mechanisms of this TKI cardiotoxicity are poorly understood, partly because the functions of many TKI target molecules in the heart are still unknown. Fms-like tyrosine kinase 3 (Flt3) is a target of TKIs used for acute myeloid leukemia treatment. Physiologically, Flt3 is mostly known as a regulator of early hematopoietic progenitor cells (Gilliland & Griffin, 2002). However, Flt3 is also expressed in non-hematopoietic tissues. We recently found that intramyocardially administered Flt3 ligand (Flt3L) confers cytoprotection in the infarcted mouse heart, thereby reducing infarct size and improving post-myocardial infarction remodeling and function (Pfister et al, 2014). Whether Flt3 also regulates cardiac progenitor cells (CPCs) has not yet been investigated.

The cardiac side population (SP) contains a pool of heterogeneous CPCs that can give rise to all cardiac lineages including cardiomyocytes, endothelial cells and vascular smooth muscle cells (Pfister et al, 2005; Noseda et al, 2015), hence contributing to cardiovascular homeostasis. Responsible for the SP phenotype are ATP-binding cassette (ABC) transporter proteins, which are regulated by PI3K/Akt signaling (Mogi et al, 2003) and can be inhibited by the Flt3-targeting TKIs quizartinib, sunitinib, ponatinib, and midostaurin (Sen et al, 2012; Wei et al, 2012; Bhullar et al, 2013; Ji et al, 2019). Flt3 activity is also a determinant of the hematopoietic SP (Chu et al, 2012). These observations give rise to the hypothesis that Flt3 participates in the regulation of cardiac SP-CPCs. Here we describe a novel role of Flt3 in the regulation of abundance, composition, and functionality of the cardiac SP.

## Results

### Absence of Flt3L is associated with a lower abundance of SP-CPCs with less contribution of CD34$^+$ cells

When comparing the SP of wild-type and flt3L-knock-out (flt3L$^{-/-}$) mice, we found diminished frequencies of SP cells in the hearts of flt3L$^{-/-}$ mice, and this was most pronounced for the cells exhibiting the highest potential to extrude Hoechst dye (tip third of the tail, Figs 1A and S1A). Freshly isolated SP cells are cell-cycle inactive and enriched in quiescent cells residing in the G0 phase of the cell cycle. We found no significant difference in the proportion of cells low in Pyronin Y expression between the wild-type and flt3L$^{-/-}$ SP (Figs 1B and S1B), suggesting that Flt3 does not affect the quiescence of SP-CPCs. The reduced SP fraction in flt3L$^{-/-}$ hearts instead supports

[1]Department of Biomedicine, University Hospital Basel and University of Basel, Basel, Switzerland  [2]Swiss Institute of Bioinformatics, Basel, Switzerland  [3]Department of Cardiology, University Hospital Basel, Basel, Switzerland

Correspondence: Gabriela.Kuster@usb.ch
*Giacomo Della Verde and Michika Mochizuki are co-first authors
†Otmar Pfister and Gabriela M Kuster are co-senior authors

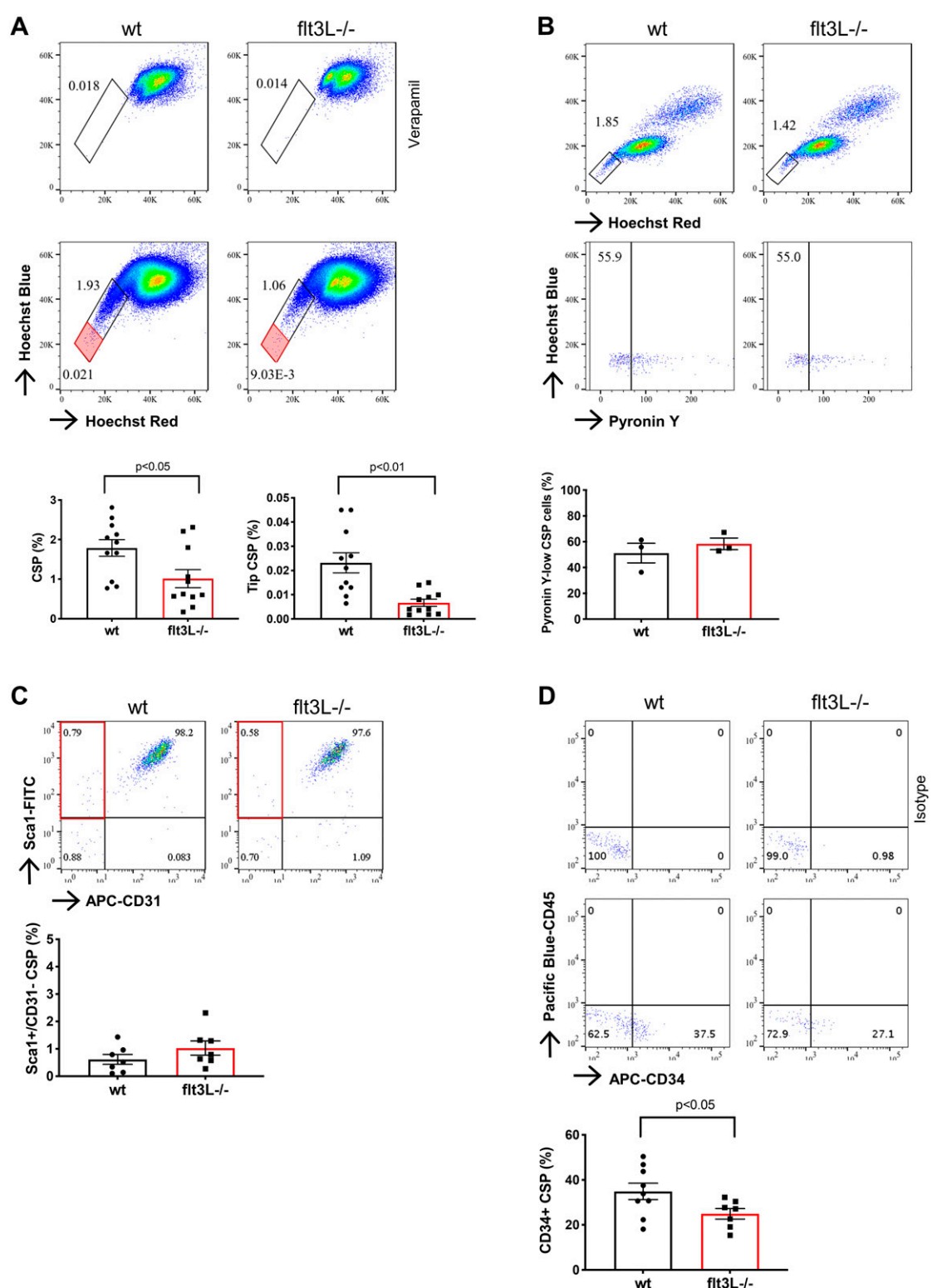

**Figure 1. Flt3L$^{-/-}$ hearts show a lower abundance and differences in the composition of the cardiac side population.**
**(A)** Percentages of cardiac side population (CSP) cells in the cardiomyocyte (CMC)- and erythrocyte-depleted cell population. Verapamil is used as negative control. The red rectangle depicts the tip third of the CSP. Wild type (wt) versus flt3L$^{-/-}$, n = 11; unpaired $t$ test. Representative flow cytometry readouts are shown. **(B)** Pyronin Y-low cells in the CSP. wt versus flt3L$^{-/-}$, n = 3; unpaired $t$ test, $P$ = 0.46. Representative flow cytometry readouts are shown. **(C)** Sca1$^+$/CD31$^-$ fraction in the CSP. wt versus flt3L$^{-/-}$, n = 7; unpaired $t$ test, $P$ = 0.21. Representative flow cytometry readouts are shown. **(D)** CD34$^+$ fraction in the CSP. wt versus flt3L$^{-/-}$, n = 9 and 7; unpaired $t$ test. Representative flow cytometry readouts are shown. Data are mean ± SEM. Each n consists of a pool of four mice.

that Flt3 signaling contributes to the SP phenotype of CPCs, which is consistent with the previously observed inhibition of ABC transporter activity under Flt3-targeting TKI therapy (Sen et al, 2012; Wei et al, 2012; Bhullar et al, 2013; Ji et al, 2019).

The Sca1⁺CD31⁻ subpopulation of the SP can be amplified in culture, while retaining tri-lineage differentiation potential (Noseda et al, 2015). This subpopulation was proportionally preserved in the SP from flt3L⁻/⁻ compared with wild-type hearts (Fig 1C), suggesting equal contribution of Sca1⁺CD31⁻ to the SP of both strains. Most CD31⁻ cells expressing the ABC transporter protein ABCG2, which is responsible for the cardiac SP phenotype, are localized in the perivascular area within the heart (Oyama et al, 2007). Because Sca1⁺/CD34⁺ and CD31⁻/CD34⁺ progenitor cells residing in the perivascular niche have been attributed vasculogenic potential (Zengin et al, 2006; Passman et al, 2008) and Flt3 is a major regulator of CD34⁺ cells in the hematopoietic system, we assessed the amount of CD34⁺ cells in the cardiac SP. Compared with wild-type, we found significantly less CD45⁻CD34⁺ cells in the SP of flt3L⁻/⁻ hearts (Fig 1D), suggesting a reduction of potentially vasculogenic cells within SP-CPCs in the absence of Flt3L.

### SP-CPCs from flt3L⁻/⁻ mice show differences in their gene expression profile

To further characterize the cardiac SP, we performed RNA sequencing on freshly isolated SP cells from flt3L⁻/⁻ and wild-type hearts. At a false discovery rate of 5%, one gene was significantly up-regulated (*Serpinh1*) and 19 were down-regulated in flt3L⁻/⁻ SP-CPCs (Table S1). These included genes with known roles in early cardiac progenitor regulation and cardiac development, such as *Ltp3* (Zhou et al, 2011), *Tenm3* (Sahara et al, 2019), and *Ror1* (Halloin et al, 2019), as well as the Wnt signaling regulators *Fzd7*, *Sfrp1* and *Bmpr1a*, which previously have been implicated in vascular network formation and homeostasis (Park et al, 2006; Foulquier et al, 2018). Gene set enrichment analysis revealed that numerous immune-related pathways as well as gene sets including extracellular matrix components were down-regulated in flt3L⁻/⁻ SP cells (Tables S2 and S3). In addition, the hallmark gene set epithelial–mesenchymal transition (Table S2) as well as Gene Ontology categories related to cardiac and coronary development and stem cell differentiation were down-regulated in flt3L⁻/⁻ SP-CPCs (Table S4). These data suggest that loss of Flt3 signaling alters the composition and genetic programming of SP-CPCs towards lower cardiovascular potential.

### Absence of flt3L leads to enhanced proliferation and less lineage commitment

Upon culturing, Sca1⁺CD31⁻ SP-CPCs enter the cell cycle and start to proliferate. They secrete Flt3L (Fig 2A) and their stimulation with exogenous Flt3L induces a dose-dependent increase in Akt phosphorylation (Fig 2B), suggesting para- and possibly autocrine responsiveness to Flt3 activation. More than half of the cells maintain a Sca1⁺CD31⁻ phenotype throughout multitudinous passages, which was comparable between the two strains (data not shown). However, when cultured in non-supplemented expansion medium (EM1), SP-CPCs isolated from flt3L⁻/⁻ hearts have an approximately fivefold higher proliferation rate compared to those

from wild-type (Fig 2Ci), whereas cell viability was not different (data not shown). These findings are consistent with the previously described increased proliferation of skeletal muscle progenitors in the absence of functional Flt3 signaling (Deasy et al, 2002; Ge et al, 2013). In contrast, when the medium was supplemented with EGF and FGF (EM2), proliferation of flt3L⁻/⁻ SP-CPCs slowed down to a comparable rate to wild-type cells (Fig 2Cii).

Further assessing the vascular smooth muscle cell commitment of Sca1⁺CD31⁻ SP-CPCs, we found lower expression of calponin in flt3L⁻/⁻ compared with wild-type SP-CPCs (Fig 2Di). However, when cultured in smooth muscle cell differentiation medium containing PDGF (SMD), calponin could be similarly induced in flt3L⁻/⁻ as in wild-type cells (Fig 2Dii). Akt phosphorylation was low in flt3L⁻/⁻ SP-CPCs when cultured in non-supplemented expansion medium (Fig 2Ei), but reconstituted to wild-type levels in the presence of EGF and FGF (Fig 2Eii) and partially reconstituted in PDGF-containing differentiation medium (Fig 2Eiii), showing that recovery of PI3K/Akt signaling paralleled the restoration of the wild-type phenotype in flt3L⁻/⁻ SP-CPCs. Taken together, these results identify deficits in CPC function in the absence of Flt3 signaling and suggest that these deficits can be at least partly compensated for by receptor tyrosine kinase-activating growth factors that signal through Akt. Consistent with this hypothesis, induction of calponin could be suppressed in wild-type SP-CPCs by inhibition of Akt (Fig 2F), which is in agreement with the previously reported PI3K/Akt–mediated unblocking of smooth muscle cell gene expression via release of forkhead transcription factor FoxO4 from the nucleus (Liu et al, 2005).

### Absence of Flt3L decreases endothelial lineage commitment and is associated with decreased cardiac microvascularization and function

Similar as vascular smooth muscle cell differentiation, endothelial cell differentiation depends on PI3K/Akt signaling (Dimmeler et al, 2001). We found decreased endothelial lineage markers in flt3L⁻/⁻ SP-CPCs. However, expression of *Vwf* and *Tie2* could be induced in vitro in VEGF-containing endothelial differentiation medium (EGM2) and vWF protein expression was not different after 3 wk of differentiation (Fig 3A and B).

To test whether the impaired vasculogenic lineage commitment of flt3L⁻/⁻ SP-CPCs was associated with structural or functional deficits at the organ level, we assessed the cardiac microvascularization and global systolic function. Flt3L⁻/⁻ hearts showed decreased capillary density (Fig 3C) and consistent impairment of systolic function in terms of left ventricular ejection fraction (Fig 3D) compared with wild-type hearts.

## Discussion

In the present study we found distinct alterations in the composition, genetic programming and functional status of SP-CPCs in the absence of Flt3L, and these were mostly related to Akt signaling. The functional deficits could at least in part be corrected through activation of other receptor tyrosine kinases in vitro, suggesting a contributory function of Flt3 to a receptor tyrosine kinase signature needed for the maintenance of a physiological cardiac SP pool.

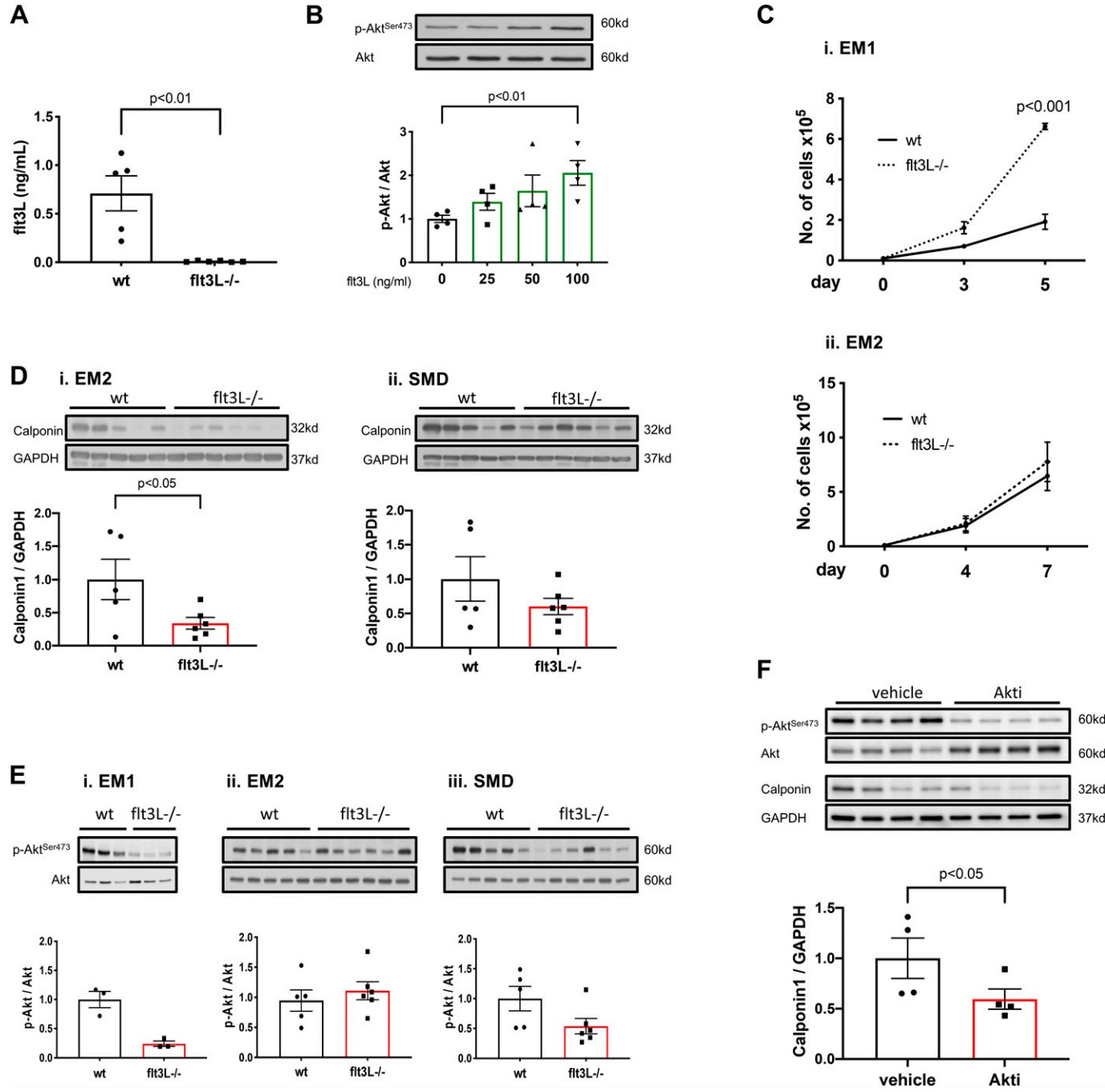

**Figure 2. Differences in in vitro behavior unveil a shift in the proliferation-differentiation balance of flt3L$^{-/-}$ side population (SP)-CPCs.**
**(A)** Flt3L quantification in supernatant from cultured wt and flt3L$^{-/-}$ Sca1$^+$/CD31$^-$ SP-CPCs after 48 h in EM2 measured by ELISA. N = 5 and 6, unpaired *t* test.
**(B)** Recombinant Flt3L-inducible signaling in wt SP-CPCs in EM1. N = 4, Kruskal–Wallis test, followed by Dunn's multiple comparison test. Representative blots are shown. **(Ci)** Sca1$^+$/CD31$^-$ SP-CPC proliferation curve in EM1, wt versus flt3L$^{-/-}$, n = 3, unpaired *t* test at final timepoint. **(ii)** SP-CPC proliferation curve in EM2, wt versus flt3L$^{-/-}$, n = 5 and 6; unpaired *t* test at final timepoint, P = 0.59. Starting cell number = 1 × 10$^4$. **(Di)** Calponin protein expression in EM2, wt versus flt3L$^{-/-}$, n = 5 and 6; unpaired *t* test. **(ii)** Calponin protein expression in smooth muscle differentiation medium (SMD) at day 7, wt versus flt3L$^{-/-}$, n = 5 and 6; unpaired *t* test, P = 0.24. Representative blots are shown. **(Ei)** Expression of phosphorylated Akt (p-Akt) in EM1, wt versus flt3L$^{-/-}$, n = 3; Mann–Whitney U test, P = 0.1. **(ii)** p-Akt expression in EM2, wt versus flt3L$^{-/-}$, n = 5 and 6; unpaired *t* test, P = 0.50. **(iii)** p-Akt expression in SMD at day 7, wt versus flt3L$^{-/-}$, n = 5 and 6; unpaired *t* test, P = 0.08. Representative blots are shown. **(F)** Calponin protein expression in wt SP-CPCs in SMD at day 7, after treatment with vehicle or 2 μM Akt inhibitor, vehicle versus treated, n = 4; ratio paired *t* test. Representative blots are shown. All data are mean ± SEM.

Our data show that the absence of Flt3L alters gene expression and diminishes the commitment of SP-CPCs towards endothelial and vascular smooth muscle cell lineages. Whether the lower frequency of potentially vasculogenic CD34$^+$ cells within flt3L$^{-/-}$ SP-CPCs is the sole reason or if and how Flt3 directly regulates gene expression and differentiation potential remains to be established.

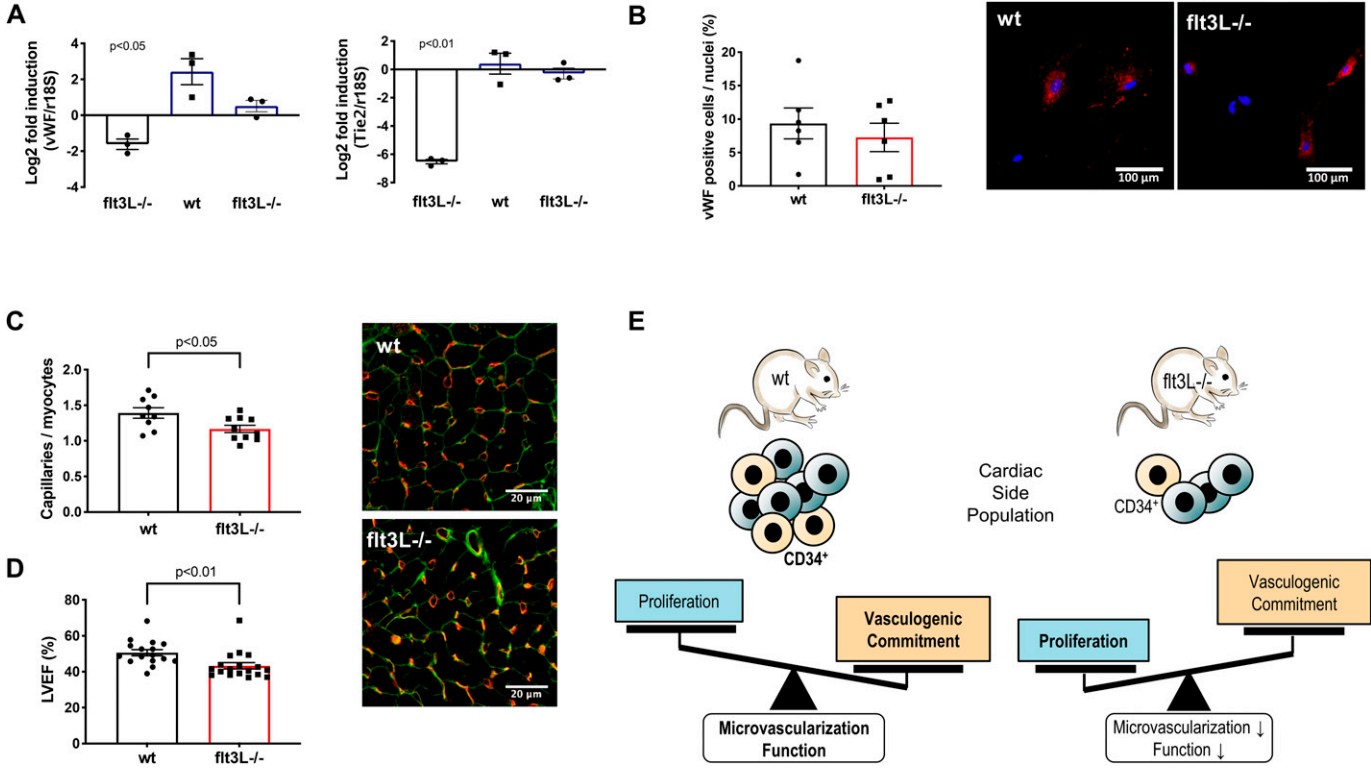

**Figure 3. The lower vasculogenic potential of flt3L$^{-/-}$ side population (SP)-CPCs reflects in a decreased microvascular density and function of flt3L$^{-/-}$ hearts.**
**(A)** Differences in endothelial marker gene expression in EM1 (black bars) and EGM2 (blue bars) versus wt EM1, n = 3, unpaired $t$ test. **(B)** Freshly isolated wt and flt3L$^{-/-}$ SP-CPCs were cultured in EGM2 for 3 wk and vWF-positive cells were quantified, n = 6, unpaired $t$ test, $P$ = 0.52. Representative pictures are shown: vWF (red), nuclei (blue). **(C)** Analysis of capillaries per myocyte in the left ventricular myocardium of wt versus flt3L$^{-/-}$ hearts, n = 9 and 10, unpaired $t$ test. Representative pictures are shown: IB4 (red), WGA (green). **(D)** Left ventricular ejection fraction (LVEF) in 12-wk-old wt and flt3L$^{-/-}$ mice, n = 16 and 18, Mann–Whitney U test. All data are given as mean ± SEM. **(E)** Proposed model of Flt3 regulation of SP-CPCs.

Interestingly, Flt3 gain-of-function as in Flt3-internal tandem duplication mutations skews the differentiation of hematopoietic blasts towards the myeloid lineage on cost of erythroid differentiation, whereas Flt3-inhibition with gilteritinib favors erythroid differentiation (Yun et al, 2019). These observations support that Flt3 has the potential to shift lineage commitment and differentiation in hematopoietic progenitor cells. Similarly, skewed differentiation can be observed in various progenitor cells during the process of aging (Schultz & Sinclair, 2016). Specifically, in aged satellite cells, myogenic differentiation declines on cost of fibrogenic differentiation and this is partially due to altered Wnt signaling (Brack et al, 2007). Therefore, lineage skewing could be a possible mechanism underlying the decreased vasculogenic commitment of SP-CPCs in the absence of intact Flt3 signaling. However, whether other lineages are favored in this process, such as cardiomyogenic or fibrogenic lineage, remains to be investigated. We also observed increased proliferation of flt3L$^{-/-}$ SP-CPCs. There exists an inverse relationship between proliferation and differentiation of progenitor cells, as these processes engage different and in part mutually exclusive gene programs. This has previously been shown in myoblasts (Skapek et al, 1995), but also applies to other types of progenitor cells. Consistent with this notion, Flt3L reduces proliferation and promotes cell cycle exit and myogenic differentiation in myoblasts (Deasy et al, 2002; Ge et al, 2013). Increased proliferation on cost of any kind of differentiation or lineage commitment in the absence of Flt3L could therefore be an alternative scenario.

The lower vasculogenic commitment of flt3L$^{-/-}$ SP-CPCs was associated with a lower capillary density and a slight, but very consistent decrease in cardiac function of flt3L$^{-/-}$ hearts. These observations are reminiscent of the decreased capillary density and cardiac function due to impaired endothelial differentiation of CPCs previously described in hearts after doxorubicin therapy (Hoch et al, 2011). Alternatively, other mechanisms could also account for the observed differences in cardiac vascularization. Specifically, Flt3 signaling may participate in the regulation of endothelial or other types of progenitor cells, which, either directly or indirectly, contribute to vascular development and maintenance (Thijssen et al, 2009). Interestingly, thalidomide was shown to protect against sunitinib-induced cardiotoxicity, possibly via PDGFR-mediated protection of pericytes (Chintalgattu et al, 2013), supporting that activation of alternative receptor tyrosine kinases can compensate for the inhibition of others, and highlighting the necessity of intact vascular architecture for cardiac homeostasis. Further studies are needed to investigate how precisely defects in Flt3 signaling affect myocardial microvascular homeostasis and to elucidate the impact of Flt3-inhibiting therapies.

### Study limitations

Although we found consistent differences between wild-type and flt3L$^{-/-}$ SP-CPCs, we could not reliably measure Flt3 receptor expression on our cells. Flt3 is rapidly internalized upon activation and cell processing may further lower surface expression. In addition, intrinsic expression of Flt3 may already be below FACS detection threshold. However, Flt3 is the only known receptor for Flt3L (Tsapogas et al, 2017) and Flt3L dose-dependently induced Akt phosphorylation in our cells, which supports the presence of functional Flt3 at least on a subset of SP-CPCs. Nevertheless, we cannot exclude that cell-extrinsic effects are partly responsible for the observed differences between wild-type and flt3L$^{-/-}$ SP-CPCs. The SP is an ex vivo phenotype and proliferation and differentiation of SP-CPCs were assessed in cultured cells. Whereas this is a well-characterized cell type in many organs including the heart (Yellamilli & van Berlo, 2016) it remains to be established to what degree our findings apply to intrinsic CPC regulation.

### Conclusion

In summary, our data suggest that Flt3 regulates the phenotype and function of murine SP-CPCs and contributes to cellular and molecular properties that are relevant for their cardiovasculogenic potential (Fig 3E). Novel insights into the functions of Flt3 in the heart may contribute to a better understanding of TKI-associated cardiotoxicity, which will ultimately benefit the cardiovascular health of patients under Flt3-targeting TKI therapy.

## Materials and Methods

Detailed information on antibodies, media composition, and primers is given in Tables S5–S7.

### Mice

Studies were conducted in 8- to 12-wk-old C57BL/6NRj (wt) and C57BL/6Flt3L$^{tml/mx}$ (flt3L$^{-/-}$) mice kindly provided by Dr Aleksandra Wodnar-Filipowicz, and originally generated and described in the laboratory of Dr Jacques Peschon (McKenna et al, 2000). Breeding and animal care were carried out by the Department of Biomedicine (DBM) Animal Facility, Hebelstrasse, Basel, Switzerland. The use of mice was in accordance with the Guide for the Care and Use of Laboratory Animals and with the Swiss Animal Protection Law and was approved by the Swiss Cantonal Authorities.

### Cardiac SP isolation and expansion, and pyronin Y staining

SP cardiac progenitor cells (CPCs) were isolated from 8 to 12 wk-old wt and flt3L$^{-/-}$ mice as previously described with some modifications (Pfister et al, 2010). In brief, mice were euthanized with 200 mg/kg pentobarbital i.p. followed by rapid excision of the heart. The heart was removed from the chest, minced with a razor blade and digested with Collagenase B 1 mg/ml (#AG 11088807001; Roche Diagnostics) supplemented with 2.5 mM CaCl$_2$ at 37°C for 25 min.

Cardiac cell suspensions were filtered with 100 and 40 $\mu$m cell strainers to eliminate undigested tissue and mature cardiomyocytes, and further treated with red blood cell lysis buffer (#420301; Biolegend) to deplete erythrocytes. Cardiomyocyte- and erythrocyte-depleted cardiac cells were suspended 1 million/ml in DMEM (#31885; Thermo Fisher Scientific) supplemented with 10% FBS (#SH30071; Hyclone) and 25 mM Hepes (#15630; Thermo Fisher Scientific) and stained with bisBenzimide H 33342 trihydrochloride (Hoechst) 5 $\mu$g/ml (#B2261; Merck) at 37°C for 90 min in the dark. Cells were washed with Hank's Buffered Salt Solution (HBSS) (#14175; Merck) and additionally incubated with Sca1-FITC (0.6 $\mu$g/$10^7$ cells) and CD31-APC (0.25 $\mu$g/$10^7$ cells) or CD34-Alexa Fluor 647 (0.25 $\mu$g/$10^7$) and CD45-Pacific blue (0.6 $\mu$g/$10^7$) at 4°C for 30 min in the dark, and with corresponding isotype controls (information on antibodies used is given in Table S5). 7-Aminoactinomycin D (7-AAD) (0.15 $\mu$g/$10^6$ cells) was added to the samples to exclude dead cells. The complete cardiac SP or the Sca1$^+$/CD31$^-$ SP-CPC fraction were sorted using BD Influx or BD FACSAriaIII SORP flow cytometer cell sorters. To quantify RNA content of SP-CPCs, Pyronin Y 2 $\mu$g/ml (#P9172; Sigma-Aldrich) was added for additional 15 min after Hoechst incubation at 4°C in the dark. Freshly isolated SP-CPCs were expanded in Expansion Medium 1 (EM1, composition given in Table S6) (Pfister et al, 2005) and Expansion Medium 2 (EM2) (Noseda et al, 2015). The medium was changed every 3 d. In vitro experiments were performed using expanded SP-CPCs from passage 7 to passage 9.

### RNA sequencing of freshly isolated cardiac SP cells

RNA from sorted SP-CPCs was isolated using the NucleoSpin RNA XS kit (Macherey Nagel). RNA from three paired sorts of wt and flt3L$^{-/-}$ SP-CPCs was used for RNA sequencing (RNA-seq), whereby each sort consisted of the cardiomyocyte- and erythrocyte-depleted cardiac cell suspension from 4 wt or 4 flt3L$^{-/-}$ mice, respectively, and each pair was processed on a different day. RNA quality was controlled on an Agilent 2100 Bioanalyzer using an RNA Pico chip (Agilent Technologies). RNA concentration was assessed using a Quantus Fluorometer (#E6150; Promega) and QuantiFluor RNA System (#E3310; Promega). Library preparations and RNA-seq were performed at the Quantitative Genomics Facility at the Department of Biosystems Science and Engineering (BSSE) of the Swiss Federal Institute of Technology Zürich in Basel, Switzerland. RNA-seq libraries were created with the SMART-Seq v4 kit from Clontech/Takara. Sequencing was performed on an Illumina NextSeq 500 sequencer, and single-end 81 bp unstranded reads were produced.

RNA-seq analysis was performed by the Bioinformatics Core Facility (Department of Biomedicine, University Hospital Basel and University of Basel). Reads were mapped to the mm10 mouse genome assembly with STAR (version 2.5.2a) (Dobin et al, 2013). We allowed at most 10 hits to the genome per read. Reads were assigned to the best mapping location, or randomly across equally best mapping locations. Using the Ensembl annotation (release 88) (Yates et al, 2020) and the qCount function from the Bioconductor package QuasR (version 1.18.0) (Gaidatzis et al, 2015), we quantified gene expression as the number of reads that started within an annotated exon of a gene. Because libraries were unstranded, only protein-coding genes were retained for the analysis. Genes with

CPM values above one in at least two samples were retained for the analysis. Between-samples normalization was performed using the trimmed mean of M-values (TMM) method (Robinson & Oshlack, 2010). Differentially expressed genes were identified using the Bioconductor package edgeR (version 3.30) (Robinson et al, 2010), using a quasi-likelihood differential expression testing framework (functions glmQLFit and glmQLFTest) (Lun et al, 2016) including the batch date as a covariate in the model. *P*-values were adjusted by controlling the false discovery rate using the Benjamini–Hochberg method (Benjamini & Hochberg, 1995). Gene set enrichment analysis was performed with the function camera (Wu & Smyth, 2012) from the limma package (version 3.44.1) using gene sets from the MSigDB Molecular Signature Database collections (MSigDB v7.0) (Liberzon et al, 2011, 2015). We filtered out sets containing less than 10 genes.

### SP-CPC expansion and culturing

Freshly isolated Sca1$^+$/CD31$^-$ SP-CPCs were cultured in Expansion Medium 1 (EM1) or 2 (EM2) (Table S6). Expanded Sca1$^+$/CD31$^-$ SP-CPCs from passages 7–9 were cultured in lineage induction medium, followed by endothelial or smooth muscle differentiation medium as described below.

### ELISA

1 × 10$^5$ Sca1$^+$/CD31$^-$ SP-CPCs were seeded in EM2 till confluence. After 48 h from the last medium change, the supernatant was collected and centrifuged using Amicon Ultra-4 Centrifugal Filter Units (#UFC801024; Merck) to achieve a retention volume of ≈150 $\mu$l. Flt3L in the supernatant was detected using R&D Systems Mouse/Rat Flt3 Ligand Quantikine ELISA kit (#MFK00; Thermo Fisher Scientific) and quantified by ELISA reader at delta 450–540 nm. Concentrated medium was used as blank.

### In vitro Flt3L dose response

1 × 10$^5$ Sca1$^+$/CD31$^-$ SP-CPCs were seeded in EM1 till confluence. Cells were then kept in 0.1% FCS overnight. Cells were treated for 5 min with Flt3L at 25, 50 and 100 ng/ml (#427-FL/CF; R&D Systems) and cell lysates were collected to quantify phosphorylated Akt by Western Blotting.

### Flow cytometry of cultured SP-CPCs

Sca1$^+$/CD31$^-$ SP-CPCs were cultured in EM2 in 75 cm$^2$ flasks till confluence. After trypsinization cells were washed with HBSS and counted using a Neubauer chamber. 5 × 10$^5$ cells were incubated for 1 h in the dark at 4°C using anti-Sca1-FITC and anti-CD31-APC antibodies (Table S5), and corresponding isotype controls. After washing with HBSS, 7-AAD was added to the samples to exclude dead cells. Samples were filtered and analyzed with BD Fortessa.

### Proliferation assay

1 × 10$^4$ Sca1$^+$/CD31$^-$ SP-CPCs were seeded (≈500 cells/cm$^2$) with EM1 and EM2. Proliferation was assessed at day 3 and 5 for EM1, and day 4 and 7 for EM2 by cell counting using a Neubauer chamber. Trypan Blue Solution (#T8154; Merck) was used to exclude dead cells.

### Smooth muscle and endothelial differentiation

5 × 10$^4$ Sca1$^+$/CD31$^-$ SP-CPCs were seeded (≈5,000 cells/cm$^2$) and kept in lineage induction medium for 48 h, and then cultured in smooth muscle differentiation medium for 7 d, up to cell lysate collection. Medium was changed every 3 d. SP-CPCs were seeded on Fibronectin (10 $\mu$g/ml)-coated cover glass and kept in lineage induction medium for 48 h, and then cultured in endothelial differentiation medium for 3 wk, up to RNA collection and immunocytochemistry. Medium was changed every 3 d.

### Western blotting

Protein samples were obtained from cell lysates using lysis buffer (#9803; Cell Signaling) containing PhosSTOP (#04-906-845-001; Merck) and Complete Protease Inhibitor Cocktail (#11-697-498-001; Merck). Protein samples were reduced with $\beta$-mercaptoethanol or dithiothreitol and heated at 95°C with shaking. Samples were loaded on 10% SDS–Page and run at 120 V. Protein samples were transferred onto polyvinylidene fluoride (PVDF) membranes previously activated with methanol. After blocking for 1 h with 4% BSA (#A7906; Merck), the membrane was incubated overnight with primary antibodies diluted in 4% BSA. GAPDH was used for normalisation. Horseradish peroxidase-conjugated antibodies (Jackson Immuno Research) were used as secondary antibodies with incubation for 1 h. Western Lightning Plus-ECL Enhanced Chemiluminescence (NEL105001EA; Perkin Elmer) was used as substrate to develop the membrane in a colorimetric or fluorescence detection. Image SXM or Fusion FX7Edge software were used to quantify protein expression. For multiple stainings, the membranes were re-blotted after stripping with Re-Blot Plus Strong Solution (#2504; Merck).

### In vitro Akt inhibition

Sca1$^+$/CD31$^-$ SP-CPCs were treated with 2 $\mu$M Akt inhibitor (#124018; Merck) (Lindsley et al, 2005; Zhao et al, 2005) from day 0 and during medium change, up to cell lysate collection for protein quantification. Dimethyl sulfoxide (#D5879; Merck) was used as negative control.

### PCR

RNA was extracted from expanded Sca1$^+$/CD31$^-$ SP-CPCs using Trizol-Chloroform (Sigma-Aldrich) or Direct-zol RNA MiniPrep Plus (#R2072; Zymo Research) and cDNA was generated using the High Capacity cDNA Reverse Transcription Kit (#4368814; Applied Biosystems). Quantitative Real Time (RT-PCR) was performed using Power SYBR Green PCR Master Mix (#4367659; Applied Biosystems). Samples were run and expression was quantified using the Fast 7500 Real Time PCR system. Primers are listed in Table S7.

### Immunocytochemistry

Sca1$^+$/CD31$^-$ SP-CPCs were washed with PBS (#20012; Thermo Fisher Scientific) and fixed in 3.7% formaldehyde solution (#F1635; Merck)

for 10 min at RT. Cells were then permeabilized with 0.1% Triton X-100 (#93420; Merck) and blocked with 10% goat serum (#50062Z; Thermo Fisher Scientific) for 1 h. Immunostaining was performed using anti–von Willebrand Factor antibody. Goat Alexa Fluor antibody was used as secondary antibody. 4′,6-diamidino-2-phenylindole, dihydrochloride (DAPI) was used to stain cell nuclei. Secondary antibody only was used as a negative control. The samples were mounted with SlowFade Antifade kit (#S2828; Thermo Fisher Scientific). The staining was visualized using Widefield Fluorescence Olympus BX63 and quantified with CellSens software.

### Echocardiography

Transthoracic echocardiography was performed using a 40 MHz probe and the Vevo 2100 Ultrasound machine (VisualSonics) equipped with a linear-array transducer (MS550) operating at a centerline frequency of 40 MHz. Echocardiography was performed in light anesthesia using isoflurane (5% for induction, 2% for maintenance). After removal of the chest hair using a depilatory agent (Nair), mice were placed on a heated platform in the supine position. Body temperature was kept around 37°C. Left parasternal short axis views at the mid papillary muscle level of the left ventricle (LV) and 2-D-guided M-mode images were used to measure LV internal diameter at end-diastole and end-systole (LVID;d and LVID;s). LV volumes in diastole (according to Teichholz: LV Vol;d = $(7.0/(2.4 + LVID;d)) \times LVID;d^3$) and systole (LV Vol;s = $(7.0/(2.4 + LVID;s)) \times LVID;s^3$) were used to calculate the LV ejection fraction (LVEF) as follows: LVEF = $100 \times (LV\ Vol;d - LV\ Vol;s)/LV\ Vol;d$.

### Perfusion and paraffin embedding

Mice were injected with 350 mM KCl (in water) intravenously (jugular vein) while under deep isoflurane anesthesia (5%) to block the heart in diastole. The mouse chest was opened with scissors. A small cut was produced on the right atrium and a 23G needle was placed longitudinally into the apex of the LV. The heart was then perfused with 15 ml of cold 4% PFA under controlled pressure (70–90 mm Hg) by a syringe pump connected to a manometer. After removing the atria, the perfused hearts were cleaned from surrounding non-heart tissue and left overnight at 4°C in 4% PFA. The hearts were then transferred to 70% ethanol and cut transversally into three pieces (apex, middle, basis) using a sterile razor blade and processed for paraffin embedding. The embedded tissue was cut into 4-$\mu$m sections on a Microm HM 340E Microtome and used for histology.

### Immunohistochemistry

The tissue was deparaffinized and rehydrated, followed by antigen retrieval using Antigen Unmasking Solution (#H3300; Vectorlabs) at 97°C for 25 min. The tissue was washed with Tris-buffered saline Tween and blocked with 10% goat serum (50-062Z; Life Technologies) for 1 h. Immunostaining was performed using FITC-labelled WGA and DyLight 594 labelled IB4. The samples were mounted with SlowFade Antifade kit. The staining was visualized using Widefield Fluorescence Olympus BX63 and quantified with CellSens software.

### Statistics

Unless otherwise indicated, data are presented as mean ± SEM. Statistical analyses were performed with GraphPad Prism version 9 software (GraphPad). Data sets were tested for Gaussian distribution using Shapiro–Wilk test and QQ plot. Normally distributed data were analyzed using $t$ test. Non-normally distributed data sets were analyzed using Mann–Whitney U test or Kruskal–Wallis test followed by Dunn's multiple comparison test for comparison of >2 groups. A $P$-value < 0.05 was considered statistically significant.

## Data Availability

RNA-seq data have been deposited at the Gene Expression Omnibus (https://www.ncbi.nlm.nih.gov/geo/) under the accession number GSE168207.

Requests for further information, resources, and reagents should be directed to the Lead Contact, Gabriela M Kuster (gabriela.kuster@usb.ch).

## Supplementary Information

## Acknowledgements

We thank Dr Aleksandra Wodnar-Filipowicz from the Department of Biomedicine (DBM), University Hospital Basel, and University of Basel, for providing flt3L$^{-/-}$ mice and scientific expertise on Flt3; the staff from the Flow Cytometry Core of the DBM for assistance in cell sorting, the staff from the Quantitative Genomics Facility at the Department of Biosystems Science and Engineering (BSSE) of the Swiss Federal Institute of Technology Zurich in Basel for RNA libraries and sequencing; Philippe Demougin from the Functional Genomics Core Facility of the University of Basel for sample preparation (RNA quantity and quality assessment); and Dr Antonius Rolink from the DBM for helpful discussions. Calculations were performed at sciCORE (http://scicore.unibas.ch/) Scientific Computing Center at the University of Basel. This study was supported by a project grant from the Swiss National Science Foundation (No 156953 to GM Kuster and O Pfister), the Nora van Meeuwen Häfliger Foundation, the Swiss Life AG Jubiläumsstiftung, the University of Basel Nachwuchsprogramm, the Swiss Heart Foundation (all to GM Kuster), and the Foundation for Cardiovascular Research, Basel, Switzerland (to GM Kuster and O Pfister).

### Author Contributions

G Della Verde: conceptualization, data curation, formal analysis, investigation, visualization, methodology, and writing—original draft, review, and editing.
M Mochizuki: conceptualization, formal analysis, validation, investigation, methodology, and writing—review and editing.
V Lorenz: data curation, formal analysis, investigation, visualization, methodology, project administration, and writing—review and editing.

J Roux: resources, data curation, software, formal analysis, validation, methodology, and writing—review and editing.

L Xu: data curation, formal analysis, investigation, visualization, and writing—review and editing.

L Ramin-Wright: investigation and writing—review and editing.

O Pfister: conceptualization, supervision, funding acquisition, validation, methodology, and writing—original draft, review, and editing.

GM Kuster: conceptualization, resources, data curation, formal analysis, supervision, funding acquisition, validation, visualization, methodology, and writing—original draft, review, and editing.

## Conflict of Interest Statement

The authors declare no conflict of interest related to this work. Outside of this work: O Pfister received consulting fees and/or speaker honoraria from AstraZeneca, Bayer, Boehringer-Ingelheim, MSD, Novartis, Pfizer, and Vifor Pharma. GM Kuster received consulting fees and speaker honoraria from Janssen-Cilag.

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
