## [Reviewer comments · Life Science Alliance]

Life Science Alliance

Fms-like tyrosine kinase 3 is a regulator of the cardiac side population in mice

Giacomo Della Verde, Michika Mochizuki, Vera Lorenz, Julien Roux, Lifen Xu, Leandra Ramin-Wright, Otmar Pfister, and Gabriela Kuster

DOI: <https://doi.org/10.26508/lsa.202101112>

Corresponding author(s): *Gabriela Kuster, University of Basel*

Review Timeline:

Submission Date:	2021-05-04
Editorial Decision:	2021-07-07
Revision Received:	2021-10-31
Editorial Decision:	2021-11-22
Revision Received:	2021-11-29
Accepted:	2021-11-30

Transaction Report:

July 7, 2021

Re: Life Science Alliance manuscript #LSA-2021-01112-T

Prof. Gabriela M Kuster
University of Basel and University Hospital Basel
Department of Biomedicine
Hebelstrasse 20
Basel 4031
Switzerland

Dear Dr. Kuster,

Thank you for submitting your manuscript entitled "Fms-like tyrosine kinase 3 is a regulator of the cardiac side population in mice" to Life Science Alliance. The manuscript was assessed by an expert reviewer, whose comments are appended to this letter. We invite you to submit a revised manuscript addressing the Reviewer comments.

When submitting the revision, please include a letter addressing the reviewer comments point by point.

Thank you for this interesting contribution to Life Science Alliance. We are looking forward to receiving your revised manuscript.

Sincerely,

-- A letter addressing the reviewer comments point by point.

B. MANUSCRIPT ORGANIZATION AND FORMATTING:

Reviewer #1 (Comments to the Authors (Required)):

The study by Della Verde et al. shows that cardiac side population progenitor cells (SP-CPCs) express and produce Flt3L. Importantly, flt3L^{-/-} mice have less SP-CPCs with less contribution of CD45-CD34⁺ endothelial progenitor cells. Upon culturing flt3L^{-/-} SP-CPCs show increased proliferation and less vasculogenic commitment. The latter depends on decreased Akt activation. Flt3L^{-/-} mouse hearts harbor a decreased microvascularisation and lower systolic function when compared to wild type hearts. These data suggest that the reduced capillarity density in flt3L^{-/-} mice depends at least in part from SP-CPCs decreased endothelial capacity secondary to flt3L knock-out. The authors fairly conclude that Flt3 regulates phenotype and function of murine SP-CPCs and contributes to cellular and molecular properties that are relevant for their cardiovascularogenic potential.

The study is well and clearly written with a solid experimental design and a set of interesting original data.

I have a few issues that I think the authors should consider to address to strengthen the presentation of their data:

1. How many SP-CPCs are Flt3 positive? This should be clearly stated and shown.
2. The author should introduce the concept of skewed differentiation potential (Schultz MB et al. Development. 2016 Jan 1;143(1):3-14. doi: 10.1242/dev.130633) that could well explain the role of Flt3L in the differentiation potential of SP-CSCs
3. The authors should be more open minded in their conclusions and clarify that decreased capillary density and cardiac function in flt3L^{-/-} mice could be also due to other cell mechanisms, such as, for example, a direct effect on endothelial progenitors (Thijssen DH, et al. Front Biosci. 2009;14:4685-702. doi: 10.2741/3560)

Response to Reviewer's comments on LSA-2021-01112-T

We thank the Reviewer for the supportive and thoughtful comments and for the great suggestions. Please find our response below:

Comment 1. How many SP-CPCs are Flt3 positive? This should be clearly stated and shown.

Response: We thank the Reviewer for this important comment and agree that it would be informative to understand the expression pattern of Flt3 on cardiac SP cells.

We analyzed Flt3 surface expression on cultured cardiac side population cells (SP-CPCs) isolated from wild-type (wt) and Flt3 receptor deficient (Flt3^{-/-}) mice. Whereas we could detect a signal on wt-SP-CPCs upon staining with anti-CD135 antibody, which was absent in isotype control samples as well as in Flt3^{-/-} SP-CPCs, we are doubtful how reliably this reflects Flt3 expression.

Figure 1. Cultured SP-CPCs isolated from wild-type (wt) and Flt3-receptor knock-out (Flt3^{-/-}) mice were trypsinized, centrifuged and counted, and resuspended in 100 μ l PBS. Cells were incubated with anti-CD135-PE (ebioscience 12-1351-81) or rat Isotype-control-PE (ebioscience 12-4321-41) for 1h on ice. After 3x washing with PBS, cells were resuspended in 400 μ l PBS and 7-AAD was added right before analysis. Cells were analysed with CyAN cytometer (Beckman Coulter). Flt3-overexpressing leukemic blasts (described in Thanasopoulou A et al., Haematologica 2014, PMC4562535) were used as positive control.

Detection of Flt3 receptor is difficult for many reasons. Rapid internalization upon activation as well as cleavage during cell processing both may lower cell surface expression in an unpredictable manner. In addition, intrinsic expression of Flt3 may already be below FACS detection threshold. For example, a role for Flt3 has previously been proposed in “Flt3-negative” long-term hematopoietic stem cells (LT-HSCs) (Chu SH et al., *Cell Stem Cell* 2012, PMC3725984). Based on these and other data, the authors argued that “Flt3 can be (...) potentially functional without abundant cell surface protein present” (Chu SH et al., *Cell Stem Cell* 2012).

Therefore, although we could detect Flt3 on the cell surface of roughly 3% of wt cardiac SP cells, which is comparable to the expression in certain hematopoietic stem cell subsets, we are insecure whether this reflects true receptor expression on our cells. We therefore prefer not to include these data in the manuscript. However, Flt3 is the only known receptor for Flt3 ligand (Flt3L). Whereas Flt3L dose-dependently induced Akt phosphorylation in wt SP-CPCs (see manuscript, Figure 2B), it failed to do so in Flt3^{-/-} SP-CPCs (Figure 2).

Figure 2. 1×10^5 Sca1⁺/CD31⁻ SP-CPCs isolated from Flt3-receptor knock-out (Flt3^{-/-}) mice were seeded in EMI and cultured until confluence. Cells were then kept at 0.1% FCS overnight, then treated for 5 min with Flt3L at 100 ng/ml (R&D Systems, #427-FL/CF) and cell lysates were collected to quantify phosphorylated (p-Akt^{Ser473}) and total Akt (Akt) by Western Blot as described (s. Supplemental Methods).

We now added a paragraph on *Study Limitations* and pointed out the fact that we do not know what the rate of receptor expression on our cells is. Consistent with this notion, we also cannot exclude that cell-extrinsic effects contribute to the described differences between wt and Flt3 ligand knock-out (FL^{-/-}) SP-CPCs.

The respective paragraph now reads:

„Although we found consistent differences between wt and flt3L^{-/-} SP-CPCs, we could not reliably measure Flt3 receptor expression on our cells. Flt3 is rapidly internalized upon activation and cell processing may further lower surface expression. In addition, intrinsic expression of Flt3 may already be below FACS detection threshold. However, Flt3 is the only

known receptor for Flt3L (Tsapogas P et al., *Int J Mol Sci* 2017), and Flt3L dose-dependently induced Akt-phosphorylation in our cells, which supports the presence of functional Flt3 at least on a subset of SP-CPCs. Nevertheless, we cannot exclude that cell-extrinsic effects are partly responsible for the observed differences between wt and flt3L^{-/-} SP-CPCs. The side population is an ex vivo phenotype and proliferation and differentiation of SP-CPCs were assessed in cultured cells. Whereas this is a well-characterized cell type in many organs including the heart (Yellamilli A and van Berlo JH, *Front Cell Dev Biol*, 2016), it remains to be established to what degree our findings apply to intrinsic CPC regulation.”

2. The author should introduce the concept of skewed differentiation potential (Schultz MB et al. *Development*. 2016 Jan 1;143(1):3-14. doi: 10.1242/dev.130633) that could well explain the role of Flt3L in the differentiation potential of SP-CSCs.

We thank the Reviewer for this excellent suggestion. The concept of skewed differentiation is now discussed in the manuscript. The respective paragraph reads as follows:

“Our data show that deficits in receptor tyrosine kinase signature signaling due to flt3L-deficiency diminish the differentiation potential of SP-CPCs towards endothelial and vascular smooth muscle cell lineages. Interestingly, Flt3 gain-of-function as in Flt3-internal tandem duplication (ITD) mutations skews differentiation of hematopoietic blasts towards the myeloid lineage on cost of erythroid differentiation, whereas Flt3-inhibition with gilteritinib favors erythroid differentiation (Yun HD et al., *Blood Adv* 2019). These observations support that Flt3 has the potential to shift lineage commitment and differentiation in hematopoietic progenitor cells. Similarly, skewed differentiation can be observed in various progenitor cells during the process of aging (Schulz MB and Sinclair DA, *Development* 2016). Specifically, in aged satellite cells, myogenic differentiation declines on cost of fibrogenic differentiation and this effect is partially due to altered Wnt signaling (Brack AS et al., *Science*, 2007). Therefore, lineage skewing could be a possible mechanism underlying the decreased vasculogenic differentiation of SP-CPCs in the absence of intact Flt3 signaling. However, whether other lineages are favored in this process, such as cardiomyogenic or fibrogenic lineage, remains to be investigated. We also observed increased proliferation of flt3L^{-/-} SP-CPCs. There exists an inverse relationship between proliferation and differentiation of progenitor cells, as these processes engage different and in part mutually exclusive gene programs. This has previously been shown in myoblasts (Skapek SX et al., *Science* 1995), but applies to other types of progenitor cells as well. Increased proliferation on cost of any kind of differentiation or lineage commitment could therefore be an alternative scenario.”

3. The authors should be more open minded in their conclusions and clarify that decreased capillary density and cardiac function in flt3L^{-/-} mice could be also due to other cell mechanisms, such as, for example, a direct effect on endothelial progenitors (Thijssen DH, et al. *Front Biosci*. 2009;14:4685-702. doi: 10.2741/3560)

We thank the Reviewer for this important notion and agree that the mechanisms underlying the decreased capillary density and cardiac function may be multifaceted. We have added comment on the possibility of an endothelial progenitor cell-based mechanism in the discussion as follows:

„...Alternatively, other mechanisms could also account for the observed differences in cardiac vascularization. Specifically, Flt3 signaling may participate in the regulation of endothelial or

other types of progenitor cells, which, either directly or indirectly, contribute to vascular development and maintenance (Thijssen DHJ et al., Front Biosci 2009).“

November 22, 2021

RE: Life Science Alliance Manuscript #LSA-2021-01112-TR

Prof. Gabriela M Kuster
University of Basel
Department of Biomedicine
Hebelstrasse 20
Basel 4031
Switzerland

Dear Dr. Kuster,

Thank you for submitting your revised manuscript entitled "Fms-like tyrosine kinase 3 is a regulator of the cardiac side population in mice". We would be happy to publish your paper in Life Science Alliance pending final revisions necessary to meet our formatting guidelines.

- please consult our manuscript preparation guidelines <https://www.life-science-alliance.org/manuscript-prep> and make sure your manuscript sections are in the correct order; -please separate the Results and Discussion section into two - 1. Results 2. Discussion, as per our formatting requirements
- please add a Summary Blurb/Alternate Abstract in our system
- Please add Figure names - Figure 1, Figure 2 and Figure 3 to the Figure files (now each figure has been listed only as "Figure" file, no number to point out particular Figure)
- please add the twitter handle of your host institute/organization as well as your own or/and one of the authors in our system
- please incorporate the Supplemental Methods and References into the matching areas in the main text. There is no character limit in these sections. The Supplemental Material should only contain the Supplemental Tables and Figure.

A. FINAL FILES:

B. MANUSCRIPT ORGANIZATION AND FORMATTING:

Sincerely,

Reviewer #1 (Comments to the Authors (Required)):

The authors have satisfactorily addressed all the issues I raised in my first revision. I have no further comments.

Response to Editor's comments on LSA-2021-01112-TR

We thank the Editor for the positive feedback. We have now reformatted the manuscript to comply with journal requirements. Furthermore, additional information on resources used for the experiments provided to the Reviewer in the previous response has been added.

Comment 1. Please consult our manuscript preparation guidelines <https://www.life-science-alliance.org/manuscript-prep> and make sure your manuscript sections are in the correct order; -please separate the Results and Discussion section into two - 1. Results 2. Discussion, as per our formatting requirements.

We now corrected the order of the manuscript sections and separated the Results and Discussion. Because the manuscript was primarily designed with the Results sectionwise merged with the Discussion, some introductory paragraphs were added to allow for easier reading of the now separated Discussion points. These paragraphs are highlighted in yellow. Please note that whereas the purpose of these paragraphs is an easier transition between topics, they do not affect the scientific interpretation of the data.

Please add a Summary Blurb/Alternate Abstract in our system.

A Summary Blurb has been added.

Please add Figure names - Figure 1, Figure 2 and Figure 3 to the Figure files (now each figure has been listed only as "Figure" file, no number to point out particular Figure)

Figure names have been added.

Please add the twitter handle of your host institute/organization as well as your own or/and one of the authors in our system.

Neither the authors nor the Department of Biomedicine have a Twitter Account.

Please incorporate the Supplemental Methods and References into the matching areas in the main text. There is no character limit in these sections. The Supplemental Material should only contain the Supplemental Tables and Figure.

The Methods (including the respective references) have now all been incorporated at the end of the main manuscript.

In addition, resource information on the mice used for additional experiments to address the Reviewer's concerns as per Response 1 has been added as follows:

For experiments provided to the Reviewer, $Flt3^{-/-}$ mice ($flk2^{-/-}$; B6.129S/SvEv- $Flt3^{tm11rl}$) originally created in and obtained from the laboratory of Dr. Lemischka (Mackarehtschian et al 1995) were used.

Reference:

*Mackarehtschian K, Hardin JD, Moore KA, Boast S, Goff SP, Lemischka IR (1995) Targeted disruption of the $flk2/flt3$ gene leads to deficiencies in primitive hematopoietic progenitors. *Immunity*. 3(1):147-161. doi:10.1016/1074-7613(95)90167-1*

November 30, 2021

RE: Life Science Alliance Manuscript #LSA-2021-01112-TRR

Prof. Gabriela M Kuster
University of Basel
Department of Biomedicine
Hebelstrasse 20
Basel 4031
Switzerland

Dear Dr. Kuster,

Thank you for submitting your Research Article entitled "Fms-like tyrosine kinase 3 is a regulator of the cardiac side population in mice". It is a pleasure to let you know that your manuscript is now accepted for publication in Life Science Alliance. Congratulations on this interesting work.

DISTRIBUTION OF MATERIALS:

Again, congratulations on a very nice paper. I hope you found the review process to be constructive and are pleased with how the manuscript was handled editorially. We look forward to future exciting submissions from your lab.

Sincerely,
